# Anxiety reduction through art therapy in women. Exploring stress regulation and executive functioning as underlying neurocognitive mechanisms

**Annemarie Abbing** [1,2]*, **Leo de Sonneville** [2,3], **Erik Baars** [1], **Daniëlle Bourne** [4], **Hanna Swaab** [2,3]

**1** Faculty of Health, University of Applied Sciences Leiden, Leiden, The Netherlands, **2** Clinical Neurodevelopmental Sciences, Faculty of Social Sciences, Leiden University, Leiden, The Netherlands, **3** Leiden Institute for Brain and Cognition, Leiden, The Netherlands, **4** Department of Psychiatry, University Medical Centre Utrecht, Utrecht, The Netherlands

* abbing.a@hsleiden.nl

**Data Availability Statement:** Abbing, Annemarie (2019), "Dataset RCT: art therapy for anxiety in

## Abstract

### Objectives

To explore possible working mechanisms of anxiety reduction in women with anxiety disorders, treated with art therapy (AT).

### Methods

A RCT comparing AT versus waiting list (WL) condition on aspects of self-regulation. Stress regulation (heart rate and heart rate variability) and executive functioning (daily behavioural and cognitive performance aspects of executive functioning (EF)) were evaluated in a pre-post design. Participants were women, aged 18–65 years with moderate to severe anxiety symptoms.

### Results

Effectiveness of AT compared to WL was demonstrated in a higher resting HRV post treatment, improvements in aspects of self-reported daily EF (emotion control, working memory, plan/organize and task monitor), but not in cognitive performance of EF, stress responsiveness and down regulation of stress. The decrease in anxiety level was associated with improvements in self-reported daily EF.

### Conclusions

AT improves resting HRV and aspects of EF, the latter was associated with art therapy-related anxiety reduction.

women", Mendeley Data, v2. http://dx.doi.org/10.17632/hnvdvbrcx9.2.

**Funding:** This study was co-funded by the Iona Foundation (www.iona.nl), Stichting AG Phoenix and the Dutch association of anthroposophic art therapy (NVKToag). These organisations had no role in study design, data collection and analysis, decision to publish, or preparation of the manuscript.

**Competing interests:** The authors have declared that no competing interests exist.

# Introduction

Every person experiences fear and anxiety in life to some degree, but for people with an anxiety disorder, the anxiety increases over time, is disproportionate to the actual danger or threat and becomes permanent [1].

The Diagnostic and Statistical Manual of Mental Disorders (DSM-5) [1] distinguishes between different types of anxiety disorders. The most common anxiety disorders are phobias, followed by social anxiety disorder (SAD), generalized anxiety disorder (GAD) and panic disorder (PD) [2]. Although the anxiety disorders may have different triggers, they share underlying features [3, 4]. An important feature that applies to all anxiety disorders is the exaggerated cognitive appraisal that is associated with the threatening situation: hyper-alert cognitive schemes lead to pathological anxiety [5].

Anxiety consists of physiological, emotional and cognitive aspects, and arises from specific personal characteristics combined with genetic, neurobiological and social factors [6], including hypersensitivity to stress and the tendency to experience strong negative emotions (nervousness, sadness, anger). According to Clark and Watson [7], anxiety is characterized by negative affect and high physiological hyperarousal. Anxiety disorders are associated with dysfunctions in self-regulation [8, 9], such as problems concerning emotion regulation, stress regulation (hyperarousal) and difficulties with cognitive regulation.

Treatment of anxiety is often aiming at changing maladaptive beliefs through cognitive behavioural therapy [10,11] and/or reducing anxiety symptoms through medication. Art therapy (AT) is also often provided in anxiety disorders, although little is known about its effectiveness. AT is a non-verbal, so-called 'experience-oriented' intervention that uses the visual arts (e.g. painting, drawing, sculpting, clay modelling) and is provided as standalone therapy or in multidisciplinary treatment programs for anxiety disorders. Outcomes of a RCT comparing three months AT with three months waiting list condition in women with anxiety disorders showed that anxiety symptoms can be reduced by AT and that there are indications that improved perceived emotion regulation plays a role in this reduction [12]. However, other aspects of self-regulation may also play a role in the reduction of anxiety symptom severity. As discussed above, individuals suffering from anxiety have also problems concerning stress regulation and difficulties with cognitive regulation, expressed as executive functioning.

## Anxiety disorders and stress regulation

Stress regulation concerns dealing with stressors. People with anxiety problems usually have stress responses that are typically accompanied by sweating, shaking, dizziness and increased heart rate [13]. These physical reactions are driven by the autonomic nervous system in the presence of a stressor. An indicator for the functioning of the autonomic nervous system is heart rate variability (HRV), which refers to the fluctuations in heart rate, also known as the variation in time between heartbeats (InterBeat Interval) [14, 15]. Parasympathetic influences from the brainstem alter the heart rate, and HRV is thus an index of cardiac flexibility [16]. HRV decreases with age and cardiac comorbidities and increases with physical activity [17]. During stress, HR increases and HRV decreases due to more sympathetic and less parasympathetic activation.

High HRV, expressing more variation in time between heart beats, is seen as an indicator for a well-functioning autonomic nervous system, which can respond to varying demands in different situations [15, 18]. HRV at rest (resting HRV) reflects self-regulation ability [19].

A review of 36 studies into the relationship between the presence of anxiety symptoms and HRV, in people with anxiety disorders compared to control groups without anxiety symptoms [20], reported that the presence of an anxiety disorder is significantly associated with lower

HRV. This was the case for people with generalized anxiety disorder (GAD), social anxiety disorder (SAD) and panic disorder (PD): lower resting HRV is associated with PD, SAD [21] and GAD [9, 21]. Lower HRV indicates autonomic inflexibility and deficits in anxiety related inhibitory processes [21]. Resting HRV is lower in individuals that worry more [22] and have high (trait) anxiety [23]. Decrease of HRV is shown in conditions of stress (e.g. time pressure), emotional strain and increased anxiety [24]. Based on these findings, heart rate and heart rate variability are considered physiological indicators of stress and anxiety [25]. An increase in heart rate may be an indication of an increased emotional state [26].

## Anxiety disorders and executive functioning

A risk factor in the development and persistence of anxiety disorders is a limitation in executive skills [27]. Executive functions (EFs) are cognitive processes that are necessary for efficient and goal-oriented behaviour. Important EF components are inhibition (the ability to stop and / or slow down behaviour (actions and thoughts)), working memory (the collection of cognitive processes that keep information temporarily accessible in order to perform mental tasks), cognitive flexibility (changing and adjusting behaviour) and planning (being able to think ahead and subdivide the process into intermediate steps towards a goal) [28, 29, 30].

Although the body of knowledge on the relationship between anxiety and EF is small, it is assumed that anxiety disorders may be the result of suboptimal cognitive regulation processes involving executive functioning. Difficulties with inhibition are found to be associated with higher levels of anxiety symptoms. Problems with EF are a risk factor for developing social anxiety [31] and adults with a generalized anxiety disorder are found to have more difficulties with inhibition than healthy controls [32].

## Present study

There is already some evidence for anxiety symptom reduction through AT and there are indications that improved perceived emotion regulation plays a role in this reduction [12]. To gain further understanding of other regulating processes that might play a role in the reduction of anxiety symptoms through AT, the question is addressed whether stress regulation and executive functioning improve as a result of AT and whether these mechanisms are related to the reduction of anxiety symptoms and may provide (further) mechanistic evidence and explanations of the AT treatment effect.

Some studies suggest a possible stress regulating effect of AT, since AT is thought to promote relaxation [33, 34]. AT is also believed to improve several aspects of executive functioning, like inhibition, because it is supposed to contribute to decrease of (a.o.) impulsivity [35, 36]. Based on these studies and expert opinions we hypothesized that AT treatment contributes to better stress regulation, because AT is thought to induce relaxation, presumably comparable to mindfulness [37, 38] and may thus have a dampening effect on the arousal. This could become visible in a lower stress response, improved down regulation after facing a stressor and as a generally lower stress level which is reflected in a higher HRV at rest.

AT may also improve cognitive regulation, which can be reflected in improvement of several aspects of executive functioning, like inhibition, sustained attention, flexibility, working memory and task monitor, because these competences are needed to perform artistic exercises and are thought to be practiced and trained during the therapy process.

## Materials and methods

In a RCT design, comparing three months AT with three months waiting list condition in women with anxiety disorders, psychophysiological outcomes (stress responsivity) as well as

daily behavioural and cognitive performance aspects of executive functioning were measured. Data were collected as part of a single-blind RCT on the effectiveness of art therapy in women with anxiety disorders. Detailed information about the study is reported elsewhere [12]. The study was approved by the Medical Ethics Committee of the Leiden University Medical Centre, the Netherlands (NL36861.018.11) and the trial was registered in the Dutch Trial Registration (https://www.trialregister.nl/trial/6661).

## Sample randomisation and intervention

Included were adult women, aged 18–65, with a primary diagnosis of generalized anxiety disorder, social phobia and/ or panic disorder (with or without agoraphobia) and with moderate to severe anxiety symptoms (scoring >7 for anxiety and/or >10 for distress on the Four Dimension Symptoms Questionnaire (4SDQ) [39]. Patients were excluded if they were aged less than 18 years or older than 65 years, had psychosis or hallucinations, alcohol or drug addiction, suicidal risk, brain pathology. All participants were recruited through posters/flyers in the practices of family doctors and by social media.

## Sample size

Based on a pre-post measurement difference in the primary outcome of 15% (considered to be a clinically relevant LWASQ total score reduction), an alpha of 0.05, a power of 0.80 and a dropout rate of 15%, the sample size was calculated: 60 participants in total (30 participants per group) (http://clincalc.com/stats/samplesize.aspx).

## Randomisation method and allocation concealment

A pre-stratification procedure was executed with four strata: use of psychotropic drugs (yes/no), moderate or severe depression symptoms (4SDQ: depression >6) (yes/ no). After pre-stratification, through block randomization (blocks of 2), participants were at random assigned to either the treatment group (AT) or the control group (WL), according to a computer-generated list (www.randomization.com). Blinding of art therapists and participants was not possible.

The study took place at 25 private art therapy practices spread throughout the Netherlands, in the period between January 2017 and March 2018.

A total of 59 women was included between January and July 2017. After pre-stratification on comorbid depression symptom level and psychopharmaceutical use, randomization resulted in an experimental group of 30 participants and a control group of 29 participants.

## Intervention

The experimental group (AT group) received 10–12 sessions AT of one hour each, during a period of three months. The specific intervention type was anthroposophic art therapy. The control group (WL) was wait listed for three months. In order to assure that the intervention tested in the study was representative for the general approach of this type of AT, only qualified and registered Dutch anthroposophic art therapists, with more than five years' experience in working with adults with anxiety, treated the participants.

## Study population

During the study, 12 patients (20%) dropped out and 47 patients (80%) completed the trial. Dropouts concerned six from AT group and six from WL group, due to lack of time (n = 3), not willing to wait for the intervention (n = 3), hospitalization or physical illness (n = 3), non-

response (n = 2) and migration (n = 1). There were no significant differences between drop-outs and completers on baseline parameters (T0), so per-protocol analysis was justified.

The participants in the two groups did not differ on key variables, including age, diagnosis, use of medication, occupation, education and familiarity with anthroposophic healthcare and outcome variables at baseline. The analysed sample of 47 patients had a mean age of 44.4 years (SD = 14,0), moderate to severe anxiety symptoms: 11.2 (SD = 4.6) and a mean duration of anxiety symptoms of 17.6 years (SD = 18.9) (range: three months—64 years (lifetime)). Medi-cation for anxiety was used by 15 participants. Multiple diagnoses applied to all participants: 25 participants met the criteria for the diagnosis GAD, 21 for social phobia and 28 for panic disorder. Ten participants suffered from (comorbid) PTSD, five participants had current comorbid depression and 16 patients experienced one or more depressive episodes prior to this study.

## Procedure

The study contained two measurement waves: pre- and post-treatment, three months apart, which consisted of online questionnaires and home-visits with physiological and neuropsy-chological measurements. The online questionnaires were completed prior to the home-visit. The protocol during the two home-visits at T0 and T1 included the measurement of stress reg-ulation and of performance-based executive functioning. The measures at the home-visits were taken by trained research assistants who were unaware of allocation. Outcome assessors who judged and analysed the results were blinded as well.

## Measures and instruments

### Measures of stress regulation

Stress regulation was measured as stress responsiveness (response and recovery), with physio-logical responses using a Biopac MP150 Acquisition System (Biopac Systems Inc., Santa Bar-bara, CA) during a stress-evoking task (Fig 1), based on the Trier Social Stress Test (TSST) [40]. The TSST is developed to measure regulation of the autonomic nervous system (ANS) during stress.

Heart rate (HR), and heart rate variability (HRV) were recorded responses. Three ECG electrodes were attached to the chest, one located near left mid-clavicular line directly below the clavicle, one near the right mid-clavicular line and one between $6^{th}$ and $7^{th}$ intercostal space on left mid-clavicular line. To stabilize the ECQ signal, a 2 Hz high pass filter and a 50 Hz notch filter were applied in *AcqKnowledge* software (version 0.3.0, Biopac System Inc). R-peaks and IBIs were visually inspected and manually corrected by two researchers (AA and DB). The corrected recordings were analysed using the PhysioData Toolbox, a MATLAB-based application [41]. For each phase (resting, stress and cool down) we calculated mean HR: mean of the continuous HR, as interpolated for the accepted Inter-Beat-Interval (IBI) data point, and $HRV_{RMSSD}$: the square root of the mean squared differences between successive IBIs (nonadjacent IBIs disregarded). HRV is the variability in the distance between R peaks, RR-interval or Inter-Beat-Interval (IBI) and refers to beat-to-beat alterations in heart rate (HR). It is a measure of both sympathetic and parasympathetic influences on the heart (Levine et al., 2016) and is related to emotional arousal [24].

The Root Mean Squared Successive Differences (RMSSD) of the HRV was used as this is the recommended measure for calculating high frequency HRV from recordings of several minutes since this measure is indicative for parasympathetic nervous system and is most com-monly used and preferred to pNN50, as it has better statistical properties [42]. Normal RMSSD mean value in a healthy population is 42ms (range 19–74) [17].

**Stress-evoking task**, consisting of three phases:

1.) *Resting phase*: relaxing for three minutes in silence, with calm piano music through headphones.

2.) *Stress induction phase*:
- Instruction: participant was asked to prepare a presentation of three minutes with a beginning, middle and end. This presentation could be about one of these topics: Dutch politics, Dutch healthcare, the American elections, climate change or the refugee problem. It was said that the presentation would be filmed and that this video would be assessed by professors from Leiden University. Attention would not only be paid to content, but also to posture and use of voice.
- Preparation: the participant could prepare for five minutes. Preparation notes were allowed, but these could not be used during the presentation.
- Presentation: the participant was asked to stand and present for three minutes. The research assistant pretended to film the presentation with a phone. However, it was not actually filmed.

3.) *Cool down phase*: the participant was told that she had done well and could relax, while sitting in silence during five minutes.

After completion of all tasks at the second home-visit, participants were debriefed and explained that they were not recorded and would not be judged.

**Fig 1. Content of the stress-evoking task.**

Because of the vulnerability of the study population, it was decided that the experiment would be stopped immediately if the subject indicated that she wanted to stop. Also, when research assistants observed or suspected a too anxious, confused or emotional state, the subject was asked if she wanted to stop the experiment, and if so, it was stopped.

## Measures of executive functioning

We used two measures of executive functioning: behavioural EF and cognitive EF. The daily behavioural EF was measured with a self-report questionnaire (BRIEF-A) and cognitive aspects of EF were measured with performance-based measures (subtests of the Amsterdam Neuropsychological Tasks (ANT)).

The Dutch version of the Behaviour Rating Inventory of Executive Function for Adults (BRIEF-A) was used to measure various aspects of daily executive functioning [43]. The BRIEF-A is a questionnaire developed for adults, and it consists of 75 items with nine clinical scales that measure various aspects of EF: four behavioural regulation scales and five metacognition scales. The behaviour regulation scales are: *inhibit*: ability to control impulses (inhibitory control) and to stop engaging in a behaviour; *shift*: cognitive flexibility, ability to move freely from one activity or situation to another; to tolerate change; to switch or alternate attention; *emotional control*: ability to regulate emotional responses appropriately; and *self-monitor*: ability to keep track of the effect of one's own behaviour on other people. The metacognition scales are: *initiate*: ability to begin an activity and to independently generate ideas or problem-solving strategies; *working memory*: ability to hold information when completing a task, when

encoding information, or when generating goals/plans in a sequential manner; *plan/organize*: ability to anticipate future events; to set goals; to develop steps; to grasp main ideas; to organize and understand the main points in written or verbal presentations; *organization of materials*: ability to put order in work, play and storage spaces (e.g. desks, lockers, backpacks, and bedrooms); and *task monitor*: ability to check work and to assess one's own performance. T-scores were calculated from the raw scores. The ranges for the clinical scales are: <60 normal; 60–65 subclinical; >65 clinical [43].

For measuring cognitive performance-based aspects of executive functioning, the Amsterdam Neuropsychological Tasks (ANT) a computer-aided assessment, was used. The ANT allows for the systematic evaluation of neuropsychological performance [44]. It has been proven to be a sensitive and valid tool in research on executive functions. Test–retest reliability and validity of the ANT-tasks are satisfactory and have been extensively described elsewhere (e.g. [45,46]).

A test battery of three tasks was chosen for this study: Baseline Speed (BS), Shifting Attention Set Visual (SSV) and Sustained Attention Dots Patterns (SAD). These tasks cover the following neuropsychological domains: alertness (intensity of attention) (BS), inhibition / mental flexibility (SSV) and sustained attention (continuous performance) (SAD). These tasks are shortly described, for detailed information including illustrations, see [47].

The BS task is a simple reaction time task, measuring of 'intensity' aspects of alertness and attention, as described by Konrad, Günther, Hanisch & Herpertz-Dahlmann [48]. On the screen a (fixation) cross is continuously projected. This cross changes unexpectedly into a square requiring the participant to press a mouse key as fast as possible, after which the square turns into a cross again, and this is repeated in 32 trials. Main outcome parameters are reaction time (RT), reflecting alertness, and the response speed stability (SD of RT), reflecting fluctuation in alertness.

The SSV task aims at measuring inhibition and attentional flexibility. The signal consists of a horizontal bar that is permanently present on which a square jumps randomly to the left or the right. In part 1, participants have to copy the movement of a green-coloured square (press left/right button on a left/right move). In part 2, participant have to do the opposite, i.e. 'mirror' the movement of a red-coloured square, and in part 3, the square randomly changes colour, requiring participants to either copy or mirror the movement of the square. The contrast between performances in part 1 and part 2 reflects inhibitory control; the contrast between performances in part 1 and part 3 (compatible responses) reflects cognitive flexibility, with larger values indicating poorer performance. Main outcome parameters of the SSV task are reaction time and accuracy (percentage of errors).

The SAD task measures sustained attention, i.e. the ability to keep performance at a certain level during a longer period of time. In this task, 600 dots patterns with 3, 4, or 5 dots appear on a computer screen in 50 series of 12 trials, each consisting of three 3-dots, 4-dots, and 5-dots patterns, presented in a pseudo random order. Participants are required to respond to 4-dots patterns by pressing a mouse key with their preferred hand ('yes'-response) and to press the other mouse key with the non-preferred hand ('no'-response) whenever 3- or 5-dots patterns are shown. Inaccurate responses, misses ('no'-responses to 4-dots) and false alarms ('yes'-responses to 3 or 5 dots) are directly followed by a beep signal. Task duration is approximately 15–20 min. Main outcome parameters are tempo (mean series completion time across 50 series), accuracy, and fluctuation in tempo. Fluctuation in tempo, the WS subject SD of 50 completion times, is taken as the primary index of sustained attention. As participants were informed about errors by a beep signal, and correct responses following an error are separately registered, post-error slowing (sensitivity to feedback) can be estimated.

## Measure of anxiety symptoms

The Dutch version of the Lehrer Woolfolk Anxiety Symptom Questionnaire (LWASQ) [49] was used to measure the anxiety level. The LWASQ is a self-report, generic anxiety instrument with 36 questions which assesses the cognitive (worry and rumination), behavioural (avoidance) and somatic (physical symptoms) aspects of anxiety. In the present study, the difference between pre- and post-measurement was used for further analysis. The reliability of the LWASQ is sufficient ($\alpha$ = .83 tot .92) and the questionnaire is suitable for the measurement of treatment effects [50].

## Statistical analysis

Statistical analyses were conducted using SPSS statistics (version 23.0) [51]. All data was checked for normal distribution using the Shapiro Wilk test, Q-Q plot and histogram.

Reasons for missing values were reported. Dropouts were compared to completers using pre-test measures on age and anxiety score, by use of independent students t-tests. No significant differences were found, so missing cases were listwise deleted and per protocol (PP) analyses were performed for all outcomes.

The following hypotheses were tested:

1. AT results in higher resting HRV

2. AT results in a lower stress response: lower increase in HR and lower decrease in HRV from resting to stress phase, respectively

3. AT results in faster recovery (HR) during cooling down

4. AT leads to improvements on self-reported daily EF

5. AT leads to improvements on performance-based EF

6. The reduction of anxiety symptoms is associated with improvements in stress recovery

7. The reduction of anxiety symptoms is associated with improvements of EF

## Evaluation of treatment effects–stress regulation

To determine if the stress paradigm did work, we evaluated the changes in HR and HRV from resting to stress and cool down phase, with all subjects of waitlist and treatment groups included. Expected were an increase in stress level at the start of the stress induction phase (shown as increase of HR and decrease of HRV) and a decrease during the cool down (shown as decrease of HR and increase of HRV). This was tested with a general linear model repeated measures analysis for variance (RM-ANOVA), with Test phase (resting, stress induction, cooling down) as within-subject (WS) factor and the pre-test HR and HRV$_{RMSSD}$ as dependent variable respectively, using a repeated contrast (resting vs. stress induction, stress induction vs. cooling down).

To examine hypothesis 1, we tested whether the therapy had influenced resting HRV (resting phase) by using a RM- ANOVA with Test moment (pre- vs. post-test) as WS factor, Group (AT vs. WL) as BS factor and resting HRV$_{RMSSD}$ as dependent variable.

Hypothesis 2 was tested using a RM-ANOVA with Test moment (pre- vs. post-test) as WS factor, and Group (AT vs. WL) as BS factor, with stress response (stress induction HR minus resting HR) as dependent variable. The same procedure was followed for HRV$_{RMSSD}$ (stress induction HRV minus resting HRV).

To examine hypothesis 3, to test whether the experimental group improved on downregulation (recovery speed), we divided the Cooling down phase in nine slices of 30 seconds each and analysed changes in HR during using a RM-ANOVA with the Cooling down phases (slice 1–9) and Test moment (pre- vs. post-test) as WS factors, Group (AT vs. WL) as BS factor, and HR as dependent variable.

### Evaluation of treatment effects–executive functioning

To examine hypothesis 4, a RM-ANOVA was performed, using Test moment (pre- vs. post-test) as WS factor and Group (AT vs. WL) as BS factor, with BRIEF-A subscale (T) scores as dependent variables, respectively.

To establish whether the study population deviated from the norm on performance-based EF measured with the ANT, we performed a MANOVA, with the z-scores of alertness (speed, fluctuation in speed), inhibition and flexibility (reaction time, error percentage) and sustained attention (tempo, fluctuation in tempo, error percentage) as dependent variables, using the intercept test for deviations from zero.

Subsequently, the treatment effects were evaluated (hypothesis 5) by means of RM-ANOVAs with Group (AT vs. WL) as BS factor and Test moment (pre- vs. post-test) as WS factor, and the outcomes on alertness (reaction time, fluctuation in reaction time) and sustained attention (tempo, fluctuation in tempo, error percentage) as dependent variables respectively. To analyse treatment-related changes in inhibition and flexibility, differences scores were computed for Inhibitory control and Cognitive flexibility, for both reaction time and error percentages. These dependent variables were analysed with RM-ANOVAs, with Group (AT vs. WL) as BS factor and Test moment (pre- vs. post-test) as WS factor.

For all analyses, a p-value of 0.05 was considered statistically significant. The effect size Partial Eta Squared ($\eta_p^2$) was calculated to assess the relevance of the effect. An effect size of 0.01–0.06 is considered a small effect, 0.06–0.14 a medium effect, and >0.14 a large effect in RM analysis [52].

### Exploration of associating factors

The sample size was calculated based on our primary aim to study the effectiveness of art therapy. The secondary aim was to explore factors influencing anxiety reduction (testing hypotheses 6 and 7). For this purpose, correlations were computed between the reduction in anxiety score and the pre-post treatment differences in EF, and aspects of stress regulation (HR and HRV). Only those aspects of EF and stress regulation that were demonstrated to improve significantly after treatment within the experimental group were entered in the correlation analysis. Only variables that significantly correlated with anxiety reduction were subsequently entered in regression analysis. Hierarchical regression analyses were planned to examine whether improvements of EF and stress recovery contributed to anxiety symptom reduction. To examine baseline predictors of anxiety symptom reduction, regression analyses were planned with the primary outcome variable (pre-post treatment difference in anxiety symptom severity) compared to baseline variables (pre-treatment values).

## Results

### Anxiety symptom severity

In a previous paper [12] that reported on this RCT, was shown that anxiety symptom severity was significantly reduced in the AT group but not in the WL group [F(1,45) = 11.49, $p$ = 0.001, $\eta_p^2$ = 0.20].

The Within-Group outcomes are presented in Table 1.

## Stress regulation

A number of measurements of participants (n = 11) could not be used due to a very distorted signal (n = 6) or uncompleted tests due to refusal or impossibility of participants to complete the paradigm (n = 5). To investigate whether there were outliers, the 1.5 x interquartile distance rule (IKA) was used. Analyses are carried out both with and without outliers. Because this did not yield different results, the outcomes of the analyses with outliers are reported.

## Evaluation of stress paradigm

The first WS contrast (resting vs. stress induction) revealed a significant effect for HR [$F(1,51)$ = 158.72, $p<0.0001$, $\eta_p^2$ = .757] and HRV [$F(1,51)$ = 5.666, $p$ = 0.021, $\eta_p^2$ = .100], respectively, with HR increasing from [mean (SD)] 69,59 (9,58) to 90,28 (15,96), and HRV decreasing from [mean(SD)] 40,02 (25,29) to 29,25 (25,25). The second contrast (stress induction vs. cool down) showed a significant effect for HR [$F(1,51)$ = 166.47, $p<0.0001$, $\eta_p^2$ = .765] and HRV [$F(1,51)$ = 6.342, $p$ = 0.015, $\eta_p^2$ = .111], with HR decreasing from [mean(SD)] 90,28 (15,96) to 70,14 (9,83) and HRV increasing from 29,25 (25,25) to 42,27 (29,49). This confirms that the stress paradigm worked: an increase of HR is shown in the stress induction phase and a decrease is shown in the cool down phase as expected. The HRV decreases during stress induction and recovers during cool down.

## Treatment effects on HRV

For $HRV_{RMSSD}$, the interaction effect Test moment*Group was trend significant [$F(1,35)$ = 3.96, p = 0.054, $\eta_p^2$ = .102], indicating that the AT group improved more than the WL group. This was further explored within the three phases. During *resting phase*, the interaction Test moment*Group was significant [$F(1,35)$ = 4.54, $p$ = 0.04, $\eta_p^2$ = .115], reflecting that the AT group had a higher HRV during the resting phase at post treatment and the WL group had a lower HRV at T1, indicating improved HRV in AT group only (Table 2 and Fig 2).

For the *stress induction phase*, no significant Test moment*Group interaction was shown ($p$ = 0.81) and in the *cool down phase*, the Test moment * Group interaction on HRV was trend significant ($p$ = 0.068): the WL group appeared to have a lower HRV at T1, compared to T0, and the AT showed an increase of HRV at T1, indicating an improvement (Table 2 and Fig 3).

## Treatment effects on stress responsivity (HR)

The RM-ANOVAs testing the treatment effects of AT on HR showed a significant main effect for Test moment, indicating an increase in HR from pre- to post treatment [$F(1,35)$ = 11.46, $p$ = 0.002, $\eta_p^2$ = 0.247], but no significant Test moment * Group interaction ($p$ = .649) (Fig 3).

Stress response was calculated as the difference in mean HR between stress induction phase and resting phase. No significant differences in stress response were found between groups ($p$ = 0.444), indicating that there were no improvements in stress responsivity.

## Treatment effect on stress recovery (HR)

Stress recovery was calculated as the difference in mean HR between stress induction phase and cool down phase. No significant differences in stress recovery were found between groups ($p$ = 0.374), indicating that there were no improvements in stress recovery.

**Table 1. *Anxiety symptom severity*: Within-Group outcomes [12].** Mean, standard deviation, 95% CIs and p-values from pre- to post-treatment (paired t-tests).

| Measure and condition | T0 Mean (SD) | T1 Mean (SD) | Mean diff (SD); 95% CI | *p*-values |
|---|---|---|---|---|
| **Anxiety score (LWASQ total** | | | | |
| AT (n = 24) | 103,21 (21.45) | 83,50 (21,36)*** | 19,71 (24,07); 95%CI: 9,55–29,87 | p = .001 |
| WL (n = 23 | 97,17 (21,66) | 97,04 (23,76) | 0,13(13,99);95%CI: -5,92–6,17 | p = .965 |
| *LWASQ somatic* | | | | |
| AT (n = 24) | 40,17 (10,51) | 33,38 (10,83)** | 6,79 (10,25); 95%CI: 2,46–11,12 | p = .004 |
| WL (n = 23 | 40,43 (10,61) | 40,74 (11,01 | -0,30 (7,49): 95%CI: -3,54–2,94 | p = .847 |
| *LWASQ behavioural* | | | | |
| AT (n = 24) | 27,04 (8,21) | 21,50 (7,85)*** | 5,54 (7,02); 95%CI: 2,58–8,51 | p = .001 |
| WL (n = 23 | 22,91 (8,64) | 22,87 (8,57 | 0.04 (4,46); 95%CI: -1,88–1,97 | p = .963 |
| *LWASQ cognitive* | | | | |
| AT (n = 24) | 36,00 (8,10) | 28,63 (7,49)*** | 7,38 (8,62); 95%CI: 3,74–11,01 | p < .0001 |
| WL (n = 23 | 33,83 (8,11 | 33,43 (9,01 | 0.39 (6,34); 95%CI:-2,35–3,14 | p = .770 |

** p<0.01

*** p<0.001

(p-values from RM-ANOVA's with Test moment (pre- vs. post-test (T0 vs T1)) as WS factor, and Group (AT vs. WL) as BS factor, with anxiety score as dependent variable)

**Table 2. Outcomes HRV and HR (stress regulation).** Mean, standard deviation at pre- and post-treatment (RM-ANOVA).

| Measure and condition | T0 Mean (SD) | T1 Mean (SD) |
|---|---|---|
| **HRV (RMSSD)** | | |
| *Resting phase* | | |
| AT (n = 19) | 37,95 (26,80) | 45,30 (40,41)* |
| WL (n = 17) | 45,33 (27,98) | 29,76 (18,99) |
| *Stress induction phase* | | |
| AT (n = 19) | 37,44 (31,24) | 33,23 (33,66) |
| WL (n = 17) | 32,22 (24,85) | 30,30 (28,54) |
| *Cool down phase* | | |
| AT (n = 19) | 40,66 (26,57) | 42,74 (28,62) |
| WL (n = 17) | 48,64 (34,39) | 32,38 (23,32) |
| **Heart rate (bpm)** | | |
| *Resting phase* | | |
| AT (n = 19) | 68,45 (7,42) | 73,00 (10,56) |
| WL (n = 1)7 | 70,97(10,93 | 77,43 (9,95) |
| *Stress induction phase* | | |
| AT (n = 19) | 87,14 (9,07) | 93,45(10,89) |
| WL (n = 17) | 90,61 (15,80) | 97,11 (14,58) |
| *Cool down phase* | | |
| AT (n = 19) | 68,50 (8,09) | 72,82(9,99) |
| WL (n = 17) | 71,06 (11,01) | 77,59 (11,43) |

*p<0.05

AT = treatment condition (3 months art therapy);

WL = waiting list condition;

T0 = pre measurement; T1 = post measurement

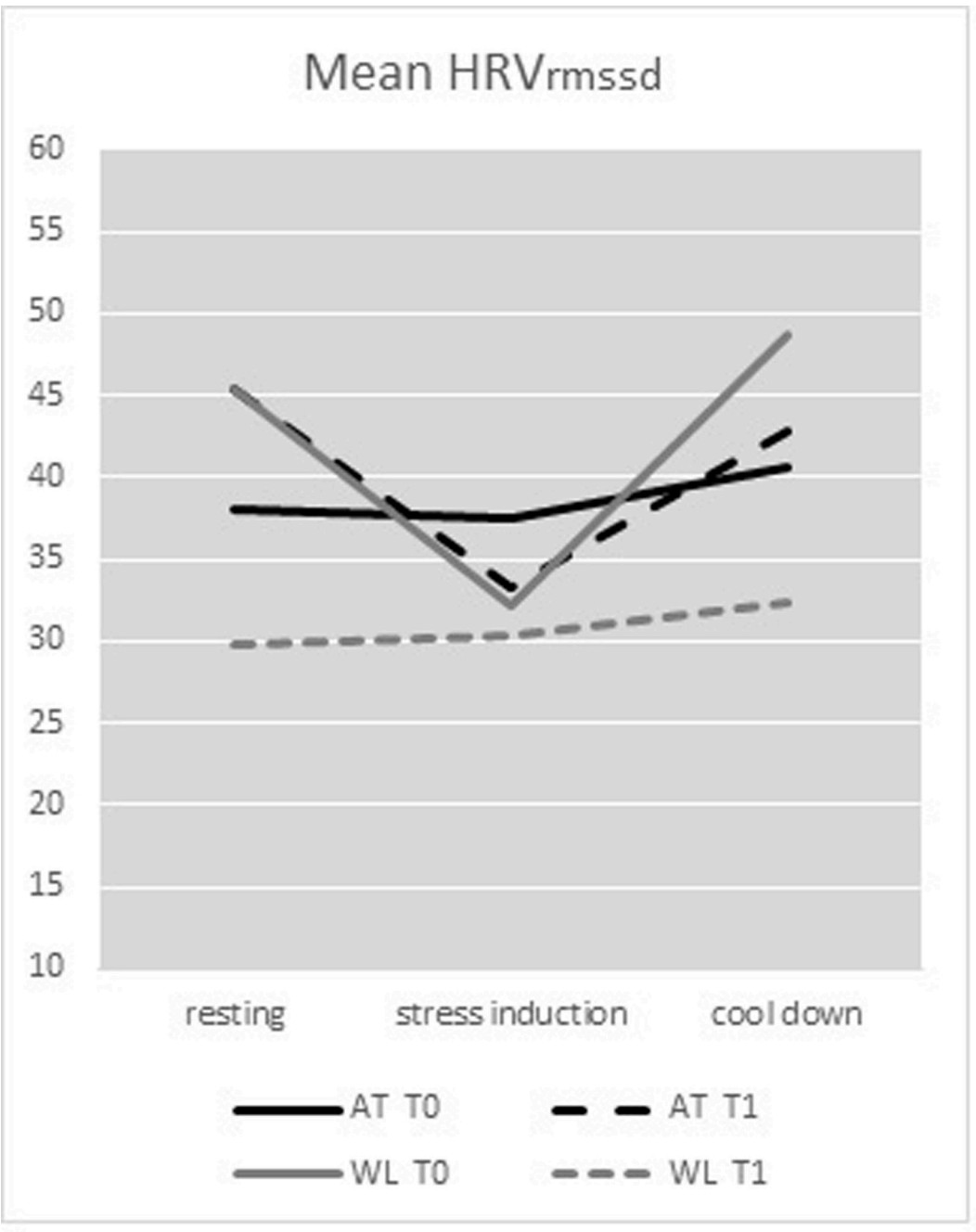

**Fig 2. Mean HRV.**

To test stress recovery speed, the cool down phase was analysed in slices of 30 seconds. No differences between the groups were observed on HR, indicating that both groups did not differ in stress recovery speed.

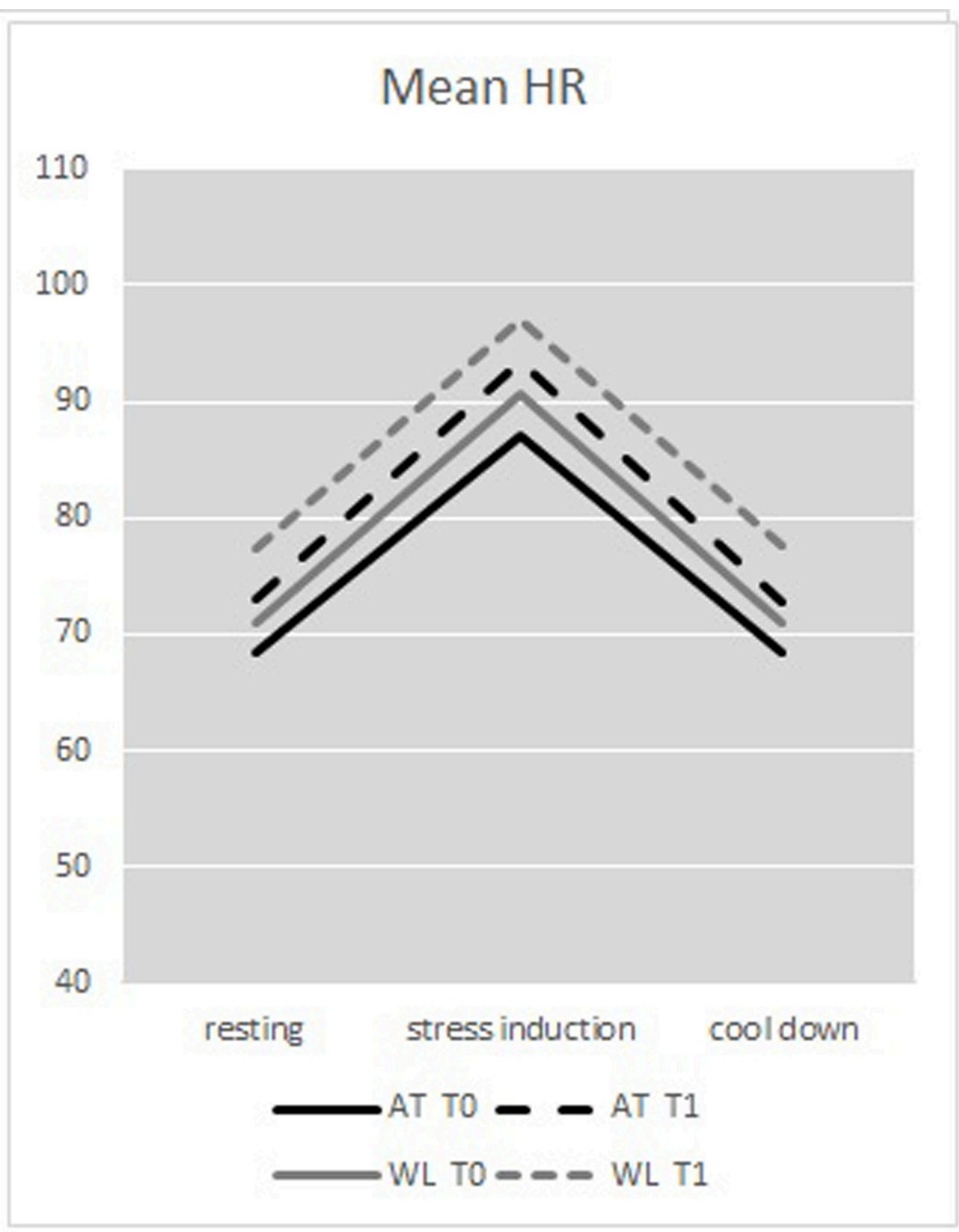

**Fig 3. Mean HR.**

### Exploration of factors contributing to anxiety symptom reduction

The only aspect of stress regulation that improved (resting HRV) did not significantly correlate to anxiety symptom reduction, so mediators and predictors were not analysed.

## Executive functioning–behavioural aspects

On the behavioural aspects of EF (BRIEF-A total score), the interaction effect Test moment*Group was significant: $F_{(1,44)} = 827$, $p = 0.006$, with a large effect size ($\eta_p^2 = .16$), showing that the total EF improved in the AT group but not in the WL group. Four of the nine subscales of the BRIEF showed significant interactions on Test moment*Group: *emotion control* [$F_{(1,44)} = 4.26$, $p = 0.045$, $\eta_p^2 = .09$]; *working memory* [$F_{(1,44)} = 5.49$, $p = 0.024$, $\eta_p^2 = .11$]; *plan/organize* [$F_{(1,44)} = 5.87$, $p = 0.020$, $\eta_p^2 = .12$] and *task monitor* [$F_{(1,44)} = 10.79$, $p = 0.002$, $\eta_p^2 = .20$], indicating that AT was effective in these domains. AT was not effective in the domains *inhibit* ($p = 0.13$), *shift* ($p = 0.24$), *self-monitor* ($p = 0.94$), *initiate* ($p = 0.66$) and *organization of materials* ($p = 0.56$) (Table 3).

## Exploration of contributing factors

Only the variables that improved significantly in the AT group (analysed with a MANOVA intercept test) were added in a regression analysis with the LWASQ difference score (pre-post treatment). These variables were *shift* (cognitive flexibility), *emotion control*, *plan/organize*, *working memory* and *task monitor*. A backward regression analysis with these variables resulted in a significant model [$F_{(3,22)} = 13,09$, $p<0.0001$, $R^2 = .674$], consisting of three subscales of the BRIEF, indicating that improvements in *emotion control* ($\beta = .364$, t = 2.54, $p = 0.020$), *plan/organize*, ($\beta = .406$, t = 2.76, $p = 0.012$), and *task monitor*, ($\beta = .319$, t = 2.21, $p = 0.039$), explained 67,4% of the variance in anxiety symptom reduction.

## Predictors

To explore possible predictors of therapy success, baseline scores of the subscales were used in a backward regression analysis in relation to anxiety symptom reduction. This resulted in a significant model [$F_{(2,20)} = 4,15$, $p = 0.031$, $R^2 = .293$] consisting of two subscales of the BRIEF, showing that higher baseline scores of *shift* ($\beta = .381$, t = 2.02, $p = 0.057$) and *organization of materials* ($\beta = .347$, t = 1.84, $p = 0.081$) led to larger reduction of anxiety symptoms, suggesting that subjects who experience many problems with these EF aspects are more likely to benefit from AT.

## Executive functioning–cognitive aspects

Intercept tests on baseline z-scores showed that some of the EF variables deviated significantly from the mean norm score, but were within normal range (-1 to 1): *fluctuation in tempo*: [mean(SD)] 0.37 (1,16); $F_{(1,44)} = 4,61$, $p = 0.037$, $\eta_p^2 = .095$; *accuracy (mistakes) in SSV1*: [mean(SD)] 0.63 (1.34); $F_{(1,44)} = 10,35$, $p = 0.002$, $\eta_p^2 = .19$; *accuracy of inhibition*: [mean (SD)] 0,80 (2,26); $F_{(1,44)} = 5,68$, $p = 0.022$, $\eta_p^2 = .114$; *accuracy of cognitive flexibility*: [mean (SD)] 0,68 (2,12); $F_{(1,44)} = 4,64$, $p = 0.037$, $\eta_p^2 = .095$.

The variable *impulsivity* falls within clinical range: [mean(SD)] 1,65 (1,65); $F_{(1,44)} = 57,07$, $p<0.0001$, $\eta_p^2 = .52$. This indicates a clinical problem (poor inhibition) in this study population.

## Treatment effects

Due to procedural errors, three cases had to be excluded from analysis, two from the AT group and one from the WL group. The RM-ANOVAs testing the treatment effects of AT on inhibition, cognitive flexibility and sustained attention showed no significant differences between experimental group and control group ($0.15<p<0.91$). Some on the tasks showed significant outcomes of test moment only, indicating a learning effect. This applied to *inhibition* (speed),

**Table 3. Outcomes BRIEF-A (executive functioning).** Mean, standard deviation, p-values and effect sizes from pre- to post-treatment (RM-ANOVA).

| Measure and condition | T0 | T1 | Time*Group | | Effect size |
|---|---|---|---|---|---|
| Executive functioning | Mean (SD) | Mean (SD) | F | *p* | (partial η²) |
| **BRIEF-A total** | | | | | |
| AT (n = 24)<br>WL (n = 23) | 67,83 (11,84)<br>61,09 (9,66) | 62,61 (9,95)<br>61,48 (9,48) | 8,27 | .006 | .16 |
| *Metacognition index* | | | | | |
| AT (n = 24)<br>WL (n = 23) | 70,61 (13,56)<br>61,35 (10,87) | 65,57 (10,54)<br>62,09 (10,89) | 6,51 | .014 | .13 |
| *Behaviour regulation index* | | | | | |
| AT (n = 24)<br>WL (n = 23) | 61,48 (10,00) 59,30 (9,27) | 56,78 (10,57)<br>58,35 (9,41) | 3,29 | .076 | .07 |
| *Inhibit* | | | | | |
| AT (n = 24)<br>WL (n = 23) | 56, 17 (11,32)<br>54,39 (8,48) | 53,87 (11,11)<br>55,13 (9,32) | 2,40 | .129 | .05 |
| *Shift* | | | | | |
| AT (n = 24)<br>WL (n = 23) | 65,59 (11,30)<br>64,36 (11,71) | 61,68 (12,09)<br>63,77 (10,65) | 1,42 | .240 | .03 |
| *Emotion Control* | | | | | |
| AT (n = 24)<br>WL (n = 23) | 61,48 (10,34)<br>58,57 (9,68) | 56,65 (10,47)<br>59,13 (10,57) | 4,26 | .045 | .09 |
| *Self-monitor* | | | | | |
| AT (n = 24)<br>WL (n = 23) | 53,00 (10.34)<br>52,30 (10,75) | 50,65 (8,89)<br>49,78 (9,20) | 0,006 | .936 | .000 |
| *Initiate* | | | | | |
| AT (n = 24)<br>WL (n = 23) | 68,48 (11,14)<br>61,09 (11,79) | 64,70 (12,12)<br>58,87 (14,28) | 0,20 | .657 | .005 |
| *Working memory* | | | | | |
| AT (n = 24)<br>WL (n = 23) | 68,35 (12,56)<br>64,22 (11,23) | 63,30 (10,59)<br>64,43 (10,74) | 5,49 | .024 | .11 |
| *Plan/Organize* | | | | | |
| AT (n = 24)<br>WL (n = 23) | 70,48 (14,15)<br>57,17 (10,92) | 66,30 (12,20)<br>59,00 (9,91) | 5,87 | .020 | .12 |
| *Task monitor* | | | | | |
| AT (n = 24)<br>WL (n = 23) | 66,78 (14,69)<br>56,30 (10,63) | 59,61 (11,75)<br>59,22 (12,02) | 10,79 | .002 | .20 |
| *Organization of materials* | | | | | |
| AT (n = 24)<br>WL (n = 23) | 62,78 (16,26)<br>58,00 (10,62) | 61,04 (14,26)<br>57,61 (11,21) | 0,35 | .555 | .008 |

AT = treatment condition (3 months art therapy); WL = waiting list condition;

T0 = pre measurement; T1 = post measurement

*flexibility* (speed) and *sustained attention* (speed and stability), but not for number of errors on the tasks and stability in speed in the inhibition and flexibility tasks. Outcomes of the tasks (mean(SD)) are presented in Table 4.

## Exploration of contributing factors

Since no significant treatment effects on cognitive performance EF were observed, associations between performance EF and anxiety symptom reduction were not analysed.

**Table 4. Outcomes of ANT tasks BS, SSV, SAD (mean(SD)).** Mean, standard deviation, p-values and effect sizes from pre- to post-treatment (RM-ANOVA).

| Measure and condition | T0 | T1 | Time*Group | | Effect size |
|---|---|---|---|---|---|
| Executive functioning | Mean (SD) | Mean (SD) | F | p | (partial η²) |
| **BS task** | | | | | |
| Alertness | | | | | |
| *Reaction time* | | | | | |
| AT (n = 23)<br>WL (n = 21) | 278 (42)<br>292 (38) | 296 (43)<br>298 (41) | 5,70 | .229 | .034 |
| *Stability* | | | | | |
| AT (n = 23)<br>WL (n = 21) | 81,43 (53,32)<br>72,38 (32,04) | 64,26 (33,65)<br>86,81 (84,35) | 1.83 | .183 | .042 |
| **SSV task** | | | | | |
| Inhibition | | | | | |
| *Reaction time (ms)* | | | | | |
| AT (n = 21)<br>WL (n = 21) | 280 (222)<br>358 (295) | 251 (158)<br>220 (159) | 2,12 | .153 | .050 |
| *Accuracy (error%)* | | | | | |
| AT (n = 21)<br>WL (n = 20) | 10,71 (22,36)<br>8,21 (9,97) | 4,29 (6,57)<br>9,40 (19,25) | 1,79 | .188 | .043 |
| Cognitive flexibility | | | | | |
| *Reaction time (ms)* | | | | | |
| AT (n = 20)<br>WL (n = 21) | 471 (254)<br>527 (285) | 360 (168)<br>385 (199) | 0,36 | .550 | .009 |
| *Accuracy (error%)* | | | | | |
| AT (n = 20)<br>WL (n = 21) | 3,75 (14,86)<br>4,88 (10,62) | 3,13 (6,97)<br>5,83 (10,96) | 0,19 | .668 | .005 |
| **SAD task** | | | | | |
| Sustained attention | | | | | |
| *Reaction time (ms)* | | | | | |
| AT (n = 22)<br>WL (n = 20) | 8,63 (1,69)<br>8,57 (1,92) | 8,20 (1,42)<br>8.04 (1,59) | 0,19 | .666 | .005 |
| *Fluctuation in tempo* | | | | | |
| AT (n = 22)<br>WL (n = 20) | 0,98 (0,41)<br>0,96 (0,60) | 0,76 (0,26)<br>0,77 (0,37) | 0,10 | .752 | .003 |
| *Accuracy (error%)* | | | | | |
| AT (n = 22)<br>WL (n = 20) | 4,14 (1,89)<br>4,15 (3,03) | 4,23 (1,75)<br>4,29 (2,78) | 0,01 | .912 | .000 |
| Behaviour regulation | | | | | |
| *Post-error slowing (ms)* | | | | | |
| AT (n = 22)<br>WL (n = 20) | 297 (277)<br>314 (250) | 181 (120)<br>223 (176) | 0,12 | .734 | .003 |

AT = treatment condition (3 months art therapy); WL = waiting list condition;

T0 = pre measurement; T1 = post measurement

## Predictors

Performance-based inhibition scores at baseline did correlate to anxiety symptom reduction (r = -.416; p = 0.043), indicating that that subjects with poorer inhibition showed a larger reduction of anxiety symptoms, suggesting these subjects are more likely to benefit from AT.

## Discussion

In this explorative study, the effects of AT on stress responsivity and executive functioning were assessed in order to further study the effectiveness of AT in adult women with anxiety and to explore possible working mechanisms of this treatment. Data were collected as part of a single-blind RCT on the effectiveness of AT in women with anxiety disorders, comparing an experimental AT treatment group and a waitlist control group [12].

Our first hypothesis that AT would contribute to better stress regulation, is partially supported. Subjects in the intervention group showed higher resting HRV after treatment, indicating a lower stress level and/or reduction of anxiety, meeting our expectation. The stress response measured after treatment was however as strong as before the treatment and no improvements in stress recovery were observed, contrary to our expectations.

Our second hypothesis that AT would result in executive functioning, was also partially supported. The results of the self-reported EF show that there were significant improvements in *emotion control*, *working memory*, *plan/organize* and *task monitor*, but the changes in AT group in the domains *inhibit*, *shift*, *self-monitor*, *initiate* and *organization of materials* were not significant compared to WL group. Regarding performance-based cognitive EF, there were no significant post treatment differences between the experimental group and the control group on Inhibition, Cognitive flexibility and Sustained attention.

The third hypothesis, that improvements in stress regulation and EF were associated with anxiety symptom reduction, was only partly substantiated. Improvements in the self-reported EF domains *emotion control*, *plan/organize* and *task monitor* were associated with anxiety symptom reduction, with an explained variance of 67,4%. Analysis of predicting factors demonstrated that lower Inhibition scores on performance EF at T0 were associated with larger reduction of anxiety symptoms, and lower self-reported *cognitive flexibility* and *organization of materials* were associated with a larger anxiety reduction.

### Interpretation and comparison to literature

The finding that improvement of resting HRV was shown in the experimental group, indicates an improved autonomic regulating ability [18, 53] and, according to the Neurovisceral Integration Model, an improved ANS regulation [54]. The higher resting HRV in the experimental group may be indicative for a lower overall stress level and can be considered an index for improved self-regulatory ability [55]. Because HRV is strongly associated with the presence of an anxiety disorder [9, 21], and HRV is positively correlated with adaptive emotion regulation, according to the Polyvagal Theory [16], the outcomes of this study substantiate our earlier finding: anxiety symptom reduction and improvement of emotion regulation [12]. Furthermore, there is neurophysiological evidence for associations between resting HRV and executive brain regions [54]. Resting HRV does not only represent overall health, but is also an index for the degree of brain flexibility concerning self-regulation processes, such as executive functions and cognitive control [56, 57].

Subjects in the experimental group showed the same stress response as before treatment and did not improve on stress recovery (down regulation). The fact that the stress response after treatment did not differ from before treatment, can have several explanations. Firstly, it is possible that subjects in the intervention group were just as sensitive/susceptible to stress induction as before treatment, leading to the preliminary conclusion that AT does not affect the direct stress response (stress responsiveness). In other studies it was shown that the stress response did not differ between healthy populations, people with intense worry and patients with GAD [58]. This implies that the stress response itself cannot easily be influenced. Secondly, the treatment period (three months) might have been to short and the number of

sessions (10–12) too little to realize significant changes in stress responsiveness. Thirdly, the Trier Social Stress task is originally developed to induce stress in healthy populations. A worry task [9] may also be suitable for this study population and may lead to other outcomes.

Another important outcome is that the treatment group experienced improvements in daily behavioural executive functioning in the domains emotion control, working memory, plan/organize and task monitor, but did not show pre-post treatment differences regarding performance-based executive functioning (Alertness, Inhibition, Cognitive flexibility and Sustained attention) compared to the control group. It is known that self-report measures are prone to a higher risk of bias / overestimation, due to positive expectations of the treated participants and to placebo effects, which are thought to account for 15% of treatment effects [59]. Positive expectations generally lead to a more positive self-evaluation of mental health [60].

A possible explanation for not finding improvements in performance EF is that the study population was not in the clinical range on this aspect (except for Inhibition), meaning that there were no major problems with EF, thus making occurrence of improvement less likely. On the other hand, the small sample size may also have compromised the outcomes. It is not unlikely that significant improvements in accuracy (error percentages) of Inhibition and Cognitive flexibility can be found in a larger study population, because these variables improved in AT group and not in WL group.

The study population showed poorer Inhibition skills compared to a healthy study population, and subjects with larger Inhibition problems showed a larger anxiety reduction. This is consistent with several studies that showed that poor behavioural inhibition is associated with anxiety and high physiological arousal [61].

## Hypotheses on working mechanisms of AT

Although there is still much unclear about the exact working mechanisms of AT, the forgoing results allow for the hypothesis that AT is effective in the treatment of anxiety symptoms due to the improvement of specific aspects of self-regulation. In our study, AT led to improvement in overall stress reduction (higher resting HRV), and the treated subjects reported improvements in several aspects of executive functioning; *emotion control*, *cognitive flexibility (shift)*, *working memory*, *plan/organize* and *task monitor*.

This hypothesis can be substantiated by the body of knowledge on anxiety reduction, which states that both higher resting HRV and improvements in EF contribute to lower anxiety levels. These improvements may be caused by the therapy. Firstly, creating visual art is linked to improved psychological resilience (i.e. stress resistance) on a neural level, due to improvements in functional connectivity of the medial parietal cortex and the praecuneus [62]. Secondly, specific skills are practiced during the artistic exercises, which are carefully chosen by the therapist. These exercises provide experiences within a safe environment and are not only intended to gain insight in emotions and responses, but are also intended to practice skills [63]. These skills may be related to aspects of executive functioning: e.g. following instructions (working memory), working autonomously on an assignment (plan/organize), tracking and evaluating own actions during the art work (task monitor), learn to interact with and adjust to the qualities of different art materials (shift), and learn to explore and regulate their emotions. This hypothetical working mechanism is substantiated by the finding that improvements of the aspects emotion control, plan/organize and task monitor contributed for 64,7% of the anxiety symptom reduction through AT.

## Strengths, limitations and generalizability

Strengths of this study are that this is the first RCT on AT for anxiety studying clinical outcomes and working mechanisms, with both self-reported and objective measurements and analyses of outcomes and influencing factors. This study is important for the scientific underpinning of AT in general and for the AT treatment of anxiety disorders specifically. Based on this study, specific hypotheses on working mechanisms can be formulated that can be tested in further research.

Although this study provides important contributions to the sparse AT literature, it does have some limitations. A first limitation is the lack of an active control group. It is therefore not possible to conclude with certainty that the observed effects are caused by therapy-specific factors. Second, the relatively small sample size may have compromised the outcomes and may have led to non-detection of significant outcomes of stress responsivity and performance EF, or to non-detection of associations between improvement of resting HRV and anxiety reduction. Also, the outcomes of the regression analyses should be handled with caution since the small sample (n = 22) that was used for these analyses compromises the generalizability of these results.

Third, cofounding and effect modifying factors such as cardiac comorbidities were not measured, which could have influenced the HRV outcomes. Fourth, the study population consisted of a specific subgroup: higher educated women, with a long duration of anxiety symptoms and probably with a specific motivation for the treatment, because the participants applied for the study. This implies that the results are not generalizable to all women with anxiety, nor to men [12].

Fifth, the study population was heterogeneous in nature, due to dimensional inclusion on anxiety symptom severity. Participants did not belong to one specific anxiety disorder classification, so conclusions on the effectiveness of art therapy for specific anxiety disorders cannot be drawn. Sixth, this study did not result in a clear substantiation of the working mechanism of art therapy, but led to preliminary hypotheses that need to be tested in further studies. Because the present study was explorative in nature and had a small sample size, straightforward analyses have been performed in order to detect promising areas for further study. Some insight into possible working mechanisms has been gained, but still many factors need to be considered before concluding on the exact working mechanism(s) of art therapy.

## Future perspectives

Outcomes of this study show that AT is a promising intervention for anxiety disorders, but studies with active controls are needed to prove efficacy and cost-effectiveness of AT. Recommendations are the testing of specific hypotheses in larger samples, testing with other objective measures and/or a different psychophysiological protocol.

Hypotheses on working mechanisms generated from this study, regarding reduction of anxiety symptom severity through improvements in self-reported daily executive functioning, should be evaluated in future studies with larger samples, and should also include mediation and moderation analyses, and should control for confounding and effect modification. Associations between outcomes on emotion regulation and executive functioning, and emotion regulation and resting HRV, need to be explored as well in order to obtain a better understanding of the route along which art therapy reduces anxiety symptom severity.

Because this explorative study contained a limited set of outcome measures, other hypotheses on working mechanisms are possible and should be studied as well as well. For instance, social regulation can be taken into account in future studies, as well as studying the effects of AT on for example work and use of pharmacotherapy for anxiety.

Other hypotheses need to be formulated more firmly and narrower, before new RCTs are designed. Hypotheses can arise in general from expert experience, practice knowledge and scientific literature. Since scientific literature on AT is in its infancy, expert experiences and practice knowledge should be used to provide input for specific clinically relevant hypotheses. Case studies are suitable for this aim [64], because they can provide basic information on therapeutic processes, are aimed at explicating expert experience and give insight in the approach that is used in this intervention (www.care-statement.org). Case studies on AT may provide insight in the route along the improvements of executive functioning and stress level are accomplished, which may further substantiate the preliminary working mechanisms found in this study. In addition, case studies can be used to describe how positively tested interventions in RCTs can be tailored to the needs of the individual patient.

## Acknowledgments

The authors thank all participants and therapists who took part in this study.

## Author Contributions

**Conceptualization:** Annemarie Abbing, Leo de Sonneville, Erik Baars, Hanna Swaab.

**Data curation:** Annemarie Abbing.

**Formal analysis:** Annemarie Abbing, Leo de Sonneville, Daniëlle Bourne.

**Funding acquisition:** Annemarie Abbing, Erik Baars.

**Investigation:** Annemarie Abbing, Daniëlle Bourne.

**Methodology:** Annemarie Abbing, Leo de Sonneville, Erik Baars, Hanna Swaab.

**Project administration:** Annemarie Abbing.

**Supervision:** Leo de Sonneville, Erik Baars, Hanna Swaab.

**Writing – original draft:** Annemarie Abbing.

**Writing – review & editing:** Leo de Sonneville, Erik Baars, Hanna Swaab.

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
