## [Decision Letter · Decision Letter 0]

11 Aug 2019

PONE-D-19-17020

Stress regulation and executive functioning in women treated for anxiety with art therapy

PLOS ONE

Dear Mrs Abbing,

Thank you for submitting your manuscript to PLOS ONE. After careful consideration, we feel that it has merit but does not fully meet PLOS ONE’s publication criteria as it currently stands. Therefore, we invite you to submit a revised version of the manuscript that addresses the points raised during the review process.

The referees bring up major points with regard to the theoretical framework, how hypotheses are derived from this, statistics, interpretation, editing and referencing. Please adjust the manuscript in line with these comments and provide detailed answers to the referees.

We would appreciate receiving your revised manuscript by Sep 25 2019 11:59PM. To enhance the reproducibility of your results, we recommend that if applicable you deposit your laboratory protocols in protocols.io, where a protocol can be assigned its own identifier (DOI) such that it can be cited independently in the future. For instructions see: http://journals.plos.org/plosone/s/submission-guidelines#loc-laboratory-protocols

We look forward to receiving your revised manuscript.

Kind regards,

Jim van Os

Academic Editor

PLOS ONE

Journal Requirements:

Reviewers' comments:

Reviewer's Responses to Questions

**Comments to the Author**

1. Is the manuscript technically sound, and do the data support the conclusions?

Reviewer #1: Partly

Reviewer #2: Partly

2. Has the statistical analysis been performed appropriately and rigorously? 

Reviewer #1: No

Reviewer #2: Yes

3. Have the authors made all data underlying the findings in their manuscript fully available?

Reviewer #1: Yes

Reviewer #2: Yes

4. Is the manuscript presented in an intelligible fashion and written in standard English?

Reviewer #1: Yes

Reviewer #2: Yes

5. Review Comments to the Author

Reviewer #1: The manuscript entitled ‘Stress regulation and executive functioning in women treated for anxiety with art therapy’ with the aim to examine the to explore possible working mechanisms of anxiety reduction in women with anxiety disorders, treated with art therapy (AT).

Comments

There numerous typographical errors due to the software where some figures with dots were presented as comma.

Materials and methods

Page 4, typo AAT.

Location of the study to be stated. The language version of all questionnaires/inventories to be clearly stated.

Page 5, the randomization method to group the subjects to be clearly stated.

Page 5, (TIST) to be placed after Trier Social Stress Test.

Sample size calculation for the study to be stated. Alternatively, power of study based on the sample size in this study could be discussed.

Page 6 rmssd to be written in capital form RMMSD. NS to be written as Nervous System.

Statistical analysis

The word stress induction or stress or stress task to be standardized where applicable.

The 7 Evaluation expectations to be replaced with the word hypothesis. The write up of the hypothesis could be further improved.

For the hypothesis 1-5, the word ‘Test moment’ to be rephrased/replaced to reflect the time. The word 'versus' or 'vs' to be standardized. pre vs post test to be added for all test moment. Likewise for AT vs WL to be consistently highlighted for the group.

Page 9 Paragraph 1 Line 3, is the cut off 0.05 for the significant level only apply for hypothesis 3 or for all analysis? If for all analysis, this has to be placed at the last section of the statistical analysis.

The sample size is small for regression analysis.

Results

Page 10 Paragraph 3 Line 9, for 'was trend significant (p=0.068), what was the cut off for the significant level in this study?

The presentation of p value using dot or 0. to be standardized throughout the manuscript e.g Page 10 paragraph 4 Line 3.

For Table 1, 2 and 3, statistical test to be denoted in the table footnote.

Discussion

The write up on the limitations could be further improved.

The referencing style in the text and references list did not conform to the journal format.

Reviewer #2: The authors describe a study on working mechanisms and predictor of art therapy outcomes among patients with anxiety disorder. This study is very valuable to this field, since there are only few RCTs on this therapy while in clinical practice art therapy is highly appreciated by clients. Nevertheless, I have several concerns on the theoretical framework, methodology, and the way conclusions are drawn. These concerns lead to my advice to major revisions before accepting this paper.

• Framework

I was confused by the descriptions on self-regulation and how this is linked to emotion and more specific anxiety. The authors mention first that emotions are part of a larger self-regulation system and next that self-regulation consist of three components (stress, cognitive, and social regulation). How does this match? What about emotion regulation? And how is this framework related to anxiety disorders? And more importantly, how will art therapy address self-regulation and from that reduce anxiety symptoms? The authors mention a bit in the conclusion on this, however this is very limited and I would expect this kind of information in the introduction. Therefore, I think that both the introduction and the conclusion could be improved by considering these interrelations in more detail.

In relation to the first comment, I was wondering on the operationalization of self-regulation. The authors used the Brief-A and three tasks of the ANT. The tasks of the ANT are cognitive tasks on aspects of executive functioning. The Brief-A is seen as a behavioral and self-reported measure on executive functioning. How does this align with the framework on self-regulation and its components? Is an emotional aspect represented in these tests? Why was no questionnaire introduced on emotion regulation?

• Methodology

Page 8: Information on reliability and validity of the LWASQ is lacking.

The authors based the analyses on the associating factors on improvements within the experimental group. In order to interpret these analyses, it would be helpful to get information on the within group effects.

Table 1 does not include any information on the statistical parameters. For the interpretation, it would be helpful to receive this information. I would expect a similar table as Table 2 and 3.

Page 13: Why were the outliers removed from the ANT tasks, while other outliers were not removed?

• Interpretation

Since there were a lot of statistical tests with only few participants, I would expect a more careful, nuanced and critical interpretation of the results. In particular, regarding the interpretation of the analyses on the associating factors. As far as I can see, these consists this of only 22 participants. Within this regard, it could be seen as a pilot to get more insight in the working mechanisms and associating factors. Regarding the interpretation of the results, see also my comments on the framework. In addition, I would expect more information on future research: what kind of hypotheses? I think there are many ideas that could be elaborated on. And how should we align RCT to case study design?

• Other comments

Introduction: the authors mention that HRV decrease with cardiac comorbidities. Was this examined in this population? And could it have any effect?

Introduction, page 4, ‘present study’: ‘…improvements is stress…’ change in ‘…improvements in stress…’

Methods, page 5: ‘…with AM and…’ needs to be ‘AT’?

Methods, page 5: what’s the difference in this study between research assistants and outcome assessors?

Methods, statistical analysis, page 8: The terminology ‘baseline HRV’ and ‘baseline HR’ is confusing, since baseline refers also to pre-test. Alternative: ‘resting HRV/HR’

Results, Page 9: HRV/HR data from eleven participant could not be used. This was due to a distorted signal or to refusal. How many were due to refusal?

6. PLOS authors have the option to publish the peer review history of their article (what does this mean?). If published, this will include your full peer review and any attached files.

Reviewer #1: No

Reviewer #2: No

---

## [Author Response · Author response to Decision Letter 0]

29 Sep 2019

Dear editor,

Thank you for considering our manuscript for publication. 

We are grateful for the detailed comments of both reviewers and we carefully made the requested adjustments.

We will discuss this below per comment.

Reviewer #1: 

Comments

There numerous typographical errors due to the software where some figures with dots were presented as comma. => We changed the comma’s into dots.

Materials and methods

Page 4, typo AAT. => We adjusted this typo.

Location of the study to be stated. The language version of all questionnaires/inventories to be clearly stated. => We added this information.

Page 5, the randomization method to group the subjects to be clearly stated. => We have now added this information

Page 5, (TIST) to be placed after Trier Social Stress Test. => Is adjusted

Sample size calculation for the study to be stated. => We have now added this information.

(Alternatively, power of study based on the sample size in this study could be discussed.)

Page 6 rmssd to be written in capital form RMMSD. => Is adjusted

NS to be written as Nervous System. => Is adjusted

Statistical analysis

The word stress induction or stress or stress task to be standardized where applicable. => Stress induction (phase) is now used throughout the manuscript 

The 7 Evaluation expectations to be replaced with the word hypothesis. => Is adjusted

The write up of the hypothesis could be further improved. => We have improved the route towards the hypotheses and the explanation on how we propose to test the hypotheses. 

For the hypothesis 1-5, the word ‘Test moment’ to be rephrased/replaced to reflect the time. => We added (pre- vs post-test) after each ‘test moment’. In the procedure section is explained that pre- and post-test were t0 and t1 measurements with three months apart.

The word 'versus' or 'vs' to be standardized. This is now adjusted.

Pre vs post test to be added for all test moment. Likewise for AT vs WL to be consistently highlighted for the group.=> Is adjusted

Page 9 Paragraph 1 Line 3, is the cut off 0.05 for the significant level only apply for hypothesis 3 or for all analysis? If for all analysis, this has to be placed at the last section of the statistical analysis. => Is now placed at the last section as suggested.

The sample size is small for regression analysis. => We agree that the regression analyses must be seen as explorative analyses only. The sample size was calculated based on our primary aim to study the effectiveness of art therapy with regard to anxiety reduction (as reported in https://doi.org/10.3389/fpsyg.2019.01203) and with regard to factors that are assumed to have an effect on anxiety level (this article). The aim of our present study was to explore factors influencing anxiety reduction, in order to make first steps towards identifying possible working mechanisms of art therapy. We presented this more clearly in the methods section, and highlighted the explorative nature in de discussion more clearly.

Results

Page 10 Paragraph 3 Line 9, for 'was trend significant (p=0.068), what was the cut off for the significant level in this study? => In the methods section, we state that a p-value <0.05 was considered significant. Because of the explorative character of this study, p-values up to 0.07 were considered trend-significant. 

The presentation of p value using dot or 0. to be standardized throughout the manuscript e.g Page 10 paragraph 4 Line 3. =>Is adjusted

For Table 1, 2 and 3, statistical test to be denoted in the table footnote. => We added this in the table headers.

Discussion

The write up on the limitations could be further improved. => This part is further extended.

The referencing style in the text and references list did not conform to the journal format. => Is adjusted

Reviewer #2: 

The authors describe a study on working mechanisms and predictor of art therapy outcomes among patients with anxiety disorder. This study is very valuable to this field, since there are only few RCTs on this therapy while in clinical practice art therapy is highly appreciated by clients. Nevertheless, I have several concerns on the theoretical framework, methodology, and the way conclusions are drawn. These concerns lead to my advice to major revisions before accepting this paper.

• Framework 

I was confused by the descriptions on self-regulation and how this is linked to emotion and more specific anxiety. The authors mention first that emotions are part of a larger self-regulation system and next that self-regulation consist of three components (stress, cognitive, and social regulation). How does this match? What about emotion regulation? And how is this framework related to anxiety disorders? And more importantly, how will art therapy address self-regulation and from that reduce anxiety symptoms? The authors mention a bit in the conclusion on this, however this is very limited and I would expect this kind of information in the introduction. Therefore, I think that both the introduction and the conclusion could be improved by considering these interrelations in more detail => We adjusted the introduction as follows: we present on the problems of individuals suffering from anxiety, discussed what is already known about art therapy and effects on anxiety reduction and emotion regulation: we have already seen that AT may lead to anxiety reduction and to improvements in emotion regulation (ER). This is the starting point for our present study. Next to ER, there are other mechanisms that play a role in anxiety: difficulties with stress regulation (arousal) and difficulties with cognitive regulation apply to individuals with anxiety. In this study, we wanted to measure effects of AT on arousal (stress regulation) and in cognitive regulation (executive functioning). AT may have a dampening effect on the arousal (stress regulation) and could possibly improve cognitive regulation, which can be reflected in improved executive functioning. We explore if these factors play a role in the reduction of anxiety. 

In relation to the first comment, I was wondering on the operationalization of self-regulation. The authors used the Brief-A and three tasks of the ANT. The tasks of the ANT are cognitive tasks on aspects of executive functioning. The Brief-A is seen as a behavioral and self-reported measure on executive functioning. How does this align with the framework on self-regulation and its components? Is an emotional aspect represented in these tests? => The BRIEF-A has a subdomain “emotion control”, but ER was not our focus in this study, since we reported on effects of AT on ER in a previous article (https://doi.org/10.3389/fpsyg.2019.01203). We explained this now more clearly in the introduction and made alterations to the introduction concerning the operationalization of the concepts.

Why was no questionnaire introduced on emotion regulation? => We included a questionnaire on ER in the RCT and the outcomes were reported in our previous article. The aim of our present study was to explore effects of AT on arousal (stress regulation) and in cognitive regulation (executive functioning) and to explore if these factors play a role in the reduction of anxiety. We acknowledge that it is also important to study relationships between ER-EF-HRV and we included this in the discussion.

• Methodology

Page 8: Information on reliability and validity of the LWASQ is lacking. => This information is now presented in the revised manuscript

The authors based the analyses on the associating factors on improvements within the experimental group. In order to interpret these analyses, it would be helpful to get information on the within group effects. => This information is now presented in the revised manuscript; detailed outcomes of the LWASQ are presented now in a new Table 1. 

Table 1 does not include any information on the statistical parameters. For the interpretation, it would be helpful to receive this information. I would expect a similar table as Table 2 and 3. =>This Table is adjusted (now Table 2, because a new Table 1 was added (outcomes of LWASQ) / anxiety). 

Page 13: Why were the outliers removed from the ANT tasks, while other outliers were not removed? => This is indeed incorrectly formulated. It concerned 3 deviations from protocol, so these cases were deleted due to procedural errors. This is now adjusted in the manuscript. 

• Interpretation

Since there were a lot of statistical tests with only few participants, I would expect a more careful, nuanced and critical interpretation of the results. In particular, regarding the interpretation of the analyses on the associating factors. As far as I can see, these consists this of only 22 participants. Within this regard, it could be seen as a pilot to get more insight in the working mechanisms and associating factors. => We agree that the regression analyses must be seen as explorative analyses only. The sample size was calculated based on our primary aim to study the effectiveness of art therapy with regard to anxiety reduction (as reported in https://doi.org/10.3389/fpsyg.2019.01203) and with regard to factors that are assumed to have an effect on anxiety level (this article). The aim of our present study was to explore factors influencing anxiety reduction, in order to make first steps towards identifying possible working mechanisms of art therapy. We presented this more clearly in the methods section, and highlighted the explorative nature in de discussion more clearly and that the outcomes of the regression analyses should be handled with caution, based on the small sample (n=22) that was used for these analyses. 

Regarding the interpretation of the results, see also my comments on the framework. In addition, I would expect more information on future research: what kind of hypotheses? I think there are many ideas that could be elaborated on. => We extended the options for future research, e.g. exploration of associations between outcomes on emotion regulation and executive functioning, and emotion regulation and resting HRV, in order to obtain a better understanding of the route along which art therapy reduces anxiety symptom severity. 

And how should we align RCT to case study design? => Two routes are possible. Case studies can be performed to identify potential working mechanisms, which can then be tested more specific on larger scale in a RCT. But case studies can also be used to describe how the intervention from the RCT can be tailored to the individual patient. We added this information in the discussion.

• Other comments

Introduction: the authors mention that HRV decrease with cardiac comorbidities. Was this examined in this population? And could it have any effect? => Subjects with a pace maker were excluded. We presumed that cardiac comorbidities were equally distributed amongst the two groups, but this was not examined. This is a limitation and is now included in the discussion. 

Introduction, page 4, ‘present study’: ‘…improvements is stress…’ change in ‘…improvements in stress…’ => Is adjusted

Methods, page 5: ‘…with AM and…’ needs to be ‘AT’? => This should be: anthroposophic healthcare; and is adjusted.

Methods, page 5: what’s the difference in this study between research assistants and outcome assessors? => Research assistants took the measures at the home visits. Outcome assessors judged the outcomes. We explained this now in the methods section. 

Methods, statistical analysis, page 8: The terminology ‘baseline HRV’ and ‘baseline HR’ is confusing, since baseline refers also to pre-test. Alternative: ‘resting HRV/HR’. => Is adjusted as suggested. 

Results, Page 9: HRV/HR data from eleven participant could not be used. This was due to a distorted signal or to refusal. How many were due to refusal? => We added this information. We also added in the method section the protocol that was used in case of refusal. 

We appreciate your time and your comments and are looking forward to your decision.

Kind regards, on behalf of all authors,

Drs. Annemarie Abbing 

University of Applied Sciences Leiden, The Netherlands

Clinical Neurodevelopmental Sciences, Leiden University, The Netherlands

Phone: + 31 71 5188 684 / E-mail: abbing.a@hsleiden.nl

---

## [Editor Report · Decision Letter 1]

31 Oct 2019

Anxiety reduction through art therapy in women. Exploring stress regulation and executive functioning as underlying neurocognitive mechanisms.

PONE-D-19-17020R1

Dear Dr. Abbing,

We are pleased to inform you that your manuscript has been judged scientifically suitable for publication and will be formally accepted for publication once it complies with all outstanding technical requirements.

With kind regards,

Jim van Os

Academic Editor

PLOS ONE
---

## [Editor Report · Acceptance letter]

12 Nov 2019

PONE-D-19-17020R1 

Anxiety reduction through art therapy in women. Exploring stress regulation and executive functioning as underlying neurocognitive mechanisms. 

Dear Dr. Abbing:

I am pleased to inform you that your manuscript has been deemed suitable for publication in PLOS ONE. Congratulations! Your manuscript is now with our production department. 

With kind regards,

on behalf of

Prof. dr. Jim van Os 

Academic Editor

PLOS ONE